

# 1 Influence of the Bermuda High on interannual variability of

# 2 summertime ozone in the Houston-Galveston-Brazoria region

**Yuxuan Wang[1,2], Beixi Jia[2], Sing-Chun Wang[1], Mark Estes[3], Lu Shen[4], Yuanyu Xie[2]**
[1]Department of Earth and Atmospheric Sciences, the University of Houston, Houston,
TX, USA
[2]Ministry of Education Key Laboratory for Earth System Modeling, Center for Earth
System Science, Tsinghua University, Beijing, China
[3]Texas Commission on Environmental Quality, Austin, TX, USA
[4]School of Engineering and Applied Sciences, Harvard University, Cambridge, MA,
USA
**ABSTRACT**
The Bermuda High (BH) quasi-permanent pressure system is the key large-scale
circulation pattern influencing summertime weather over the eastern and southern
US. Here we developed a multiple linear regression (MLR) model to characterize the
effect of the BH on year-to-year changes of monthly-mean maximum daily 8-hour
average (MDA8) ozone in the Houston-Galveston-Brazoria (HGB) metropolitan
region during June, July and August (JJA). The BH indicators include the longitude of
the BH western edge (BH-Lon), and the BH intensity index (BHI) defined as the
pressure gradient along its western edge. Both BH-Lon and BHI are selected by MLR
as significant predictors ($p < 0.05$) of the interannual (1990-2015) variability of the
HGB-mean ozone throughout JJA, while local-scale meridional wind speed is selected
as an additional predictor for August only. Local-scale temperature and zonal wind





speed are not identified as important factors for any summer month. The best-fit
MLR model can explain 61%-72% of the interannual variability of the HGB-mean
summertime ozone over 1990-2015 and shows good performance in cross-validation
($R^2$ higher than 0.48). The BH-Lon is the most important factor, which alone explains
38%-48% of such variability. The location and strength of the Bermuda High appears
to control whether or not low-ozone maritime air from the Gulf of Mexico can enter
southeastern Texas and affect air quality. This mechanism also applies to other
coastal urban regions along the Gulf Coast (e.g. New Orleans, LA; Mobile, AL; and
Pensacola, FL), suggesting that the BH circulation pattern can affect surface ozone
variability through a large portion of the Gulf Coast.
**Keyword**: Bermuda High, ozone, Houston, Gulf Coast, multiple linear regression
**1.  Introduction**
Surface ozone, as an important air pollutant, has significant adverse impacts on
both public health and agriculture (NRC, 1991). Ozone is produced in the
troposphere by photochemical oxidation of carbon monoxide (CO) and volatile
organic carbon (VOCs), initiated by reaction with hydroxyl radicals (OH) in the
presence of nitrogen oxides ($NO_x$). Surface ozone is influenced not only by emissions
of its precursors but also by meteorological conditions (e.g. Jacob and Winner 2009).
Large-scale circulation patterns can lead to local meteorological conditions that are
favorable for ozone episodes, such as high temperatures, low wind speeds, clear





skies, and stagnation (Nielsen-Gammon et al., 2005; Ngan and Byun, 2011; Pearce et
al., 2011; Psilogloue et al., 2013; Pugliese et al., 2014). Previous studies have
demonstrated certain associations between large-scale circulations and surface
ozone concentrations over the US (e.g. Darby, 2005; Rappenglück et al., 2008; Lin et
al., 2012; Zhu and Liang, 2013; Shen et al., 2015; Lin et al., 2015). For example,
surface ozone in the western US is affected by mid-latitude cyclones that transport
Asian pollutions eastward across the Pacific (Lin et al., 2012) and late-spring
stratospheric intrusions occurring more frequently following strong La Nina winters
(Lin et al., 2015). In the Midwest and Northeast US, polar jet frequency is a good
indicator for the interannual variability of surface ozone in summer (Shen et al.,

2015).

The Bermuda High (BH), a quasi-permanent system located over the North

Atlantic Ocean (Davis et al., 1997), is the key large-scale circulation pattern that
influences the regional climate over the eastern and southern US in summer (Li et al.,
2012; Zhu and Liang, 2013; Hegarty et al., 2007; Hogrefe et al., 2004). The BH
circulation pattern influences ozone air quality in the US through two mechanisms.
First, as the BH shifts westward from late spring to summer, it places the eastern US
under the high pressures, resulting in meteorological conditions favorable for local
production of ozone, such as high temperatures, clear skies, and stagnation. A
number of studies have shown that high ozone concentrations easily occur over
large parts of the Northeast under the BH pressure pattern (Eder et al., 1993; Fiore
et al., 2003; Hogrefe et al., 2004). Second, the southerly flows at the western edge of



the BH bring clean marine air from the Gulf of Mexico to the southern Great Plains
(Higgins et al. 1997). This maritime inflow has lower concentrations of ozone
compared with the continental air it replaces, thus responsible for low ozone
background over much of the Gulf States in summer (Langford et al., 2009; Ngan and
Byun et al., 2011). These two mechanisms have been illustrated by Zhu and Liang
(2013). Using observational data, they found positive correlations over the Northeast
between maximum daily 8h average (MDA8) surface ozone in summer and the
intensity of the BH on the interannual time scale due to the first mechanism, but
negative correlations over the South-Central US due to the second mechanism. Shen
et al. (2015) further suggested that the location of the BH western edge has an
influence on the summer mean MDA8 ozone in the Southeast.

The Houston-Galveston-Brazoria (HGB) area is a major metropolitan area on the

Gulf coast located near the western edge of the BH in summer (Figure 1). The HGB
region was classified as a "marginal" nonattainment zone for ozone by the U.S.
Environmental Protection Agency (EPA) under the 2008 standard (TCEQ, 2012),
although mean and peak ozone of the HGB area has decreased significantly during
the past decades due to control of anthropogenic emissions (Berlin et al. 2013). A
number of studies have demonstrated the importance of meteorology and
circulation patterns on ozone over the HGB (Ngan et al., 2011; Rappenglück et al.,
2008; Pakalapati et al., 2009; Haman et al., 2014), focusing predominantly on a
typical year or episodic high ozone cases, such as those observed during the Texas
Air Quality Study-II (Rappenglück et al., 2008; Bridget et al., 2009). Doppler lidar



measurements of the strength and direction of the nocturnal low level jet (LLJ)
during the TexAQS 2006 field campaign (Tucker et al., 2009) found a relationship
between the nocturnal LLJ and ozone concentrations on the next day. A strong
southerly nocturnal LLJ was linked to the influence of the BH. Given the evidence
established by prior investigations on the overall influence of the BH on summertime
ozone over the southern US (e.g. Zhu and Liang 2013; Shen et al., 2015), we
hypothesize that the large-scale circulation patterns associated with the BH play a
key role in driving the year-to-year change in surface ozone over the HGB. In this
study we will test this hypothesis by examining the statistical relationship between
the variability of the BH and MDA8 ozone over the HGB during June, July, and August
(JJA), a time period when the BH is located closer to North America and exerts large
influences to circulation patterns over the HGB.

**2. Data and Methods**
**2.1. Ozone observations**
The HGB region is delineated by longitude from -94.5°W to -96.0°W, and by
latitude from 28.5°N to 30.5°N (blue box in Figure 1a). Surface ozone concentrations
over the HGB have been routinely monitored at continuous ambient monitoring
stations (CAMSs) maintained by the Texas Commission on Environmental Quality
(TCEQ), the City of Houston, and Harris County. Observational records of surface
ozone in JJA from 1990 to 2015 were obtained from the EPA AirData website



(http://www3.epa.gov/airquality/airdata/ad_data_daily.html). Ozone observations
from 28 ozone CAMS sites in the HGB region that have data records longer than 10
years were used for analysis. Figure 1b displays the site distributions and long-term
mean ozone in JJA at each site. The site locations and operation time periods are
provided in the supplementary material (Table S1). The number of CAMS sites
increases to 16 in 1998 and 21 in 2004. There are 8 sites with continuous
observations from 1990 to present. The overall data coverage at each selected site is
99% during its operation period, except the Houston East site that has ozone
observations since 1990 but no data during 1993-1995. In what follows, all the ozone
data and related discussions are MDA8 ozone and the correlation coefficient, $r$,
reported in this study is the Pearson correlation coefficient unless stated otherwise.
As shown in Figure 1b, the sites in Galveston and Brazoria counties have lower
ozone concentrations than the sites in the Houston region, due in part to lower local
emissions. Since the scope is on synoptic scale circulation patterns, we focus on the
HGB-mean ozone rather than ozone at individual sites. To calculate the HGB-mean
ozone for a given month, we averaged ozone observations from all eligible sites
(all-site average) and compared that with the average of only those sites with
continuous observations since 1990 (continuous-site average). The difference in the
HGB-mean ozone calculated from the two averaging approaches is within 5 ppbv and
diminishes in the 2000s as the sites number increases (Fig 2a). To remove the
influence of the decreasing trend in anthropogenic emissions on the interannual
variability of ozone, the HGB-mean ozone of each month (JJA) was de-trended by



subtracting the 7-year moving average of the corresponding month. The de-trending
process further decreases the difference in the HGB-mean ozone calculated from the
two averaging approaches (Fig 2b). Therefore we use the all-site average to present
the HGB-mean ozone. The time series of the de-trended HGB-mean ozone is from
1993 to 2012.
**2.2.   Meteorological data**

The meteorological data consists of the geopotential height at 850hPa and sea

level pressure (SLP) from the National Centers for Environmental Prediction (NCEP)
Reanalysis 1 with a spatial resolution of $2.5^{o} \times 2.5^{o}$ (Kalnay et al., 1996), which were
used to derive the BH indices described below. We adopted 2-meter temperature (T),
zonal (U) and meridional (V) component of wind on 850hPa from the European
Centre for Medium-Range Weather Forecasts (ECMWF) Interim reanalysis with a
finer spatial resolution of $0.5^{o} \times 0.5^{o}$. These data were used to calculate the local-scale
meteorological parameters, including the HGB-mean temperature and 850 hPa
winds (U and V).
**2.3.   BH indices**

Figure 3 illustrates the mean BH circulation patterns from June to August. The

HGB region is located to the west of the BH, thus under strong southerly winds from
June to August. The intensity of the BH peaks in July. As the BH intensifies from June
to July, its west edge moves westward toward the North America continent; when it
weakens from July to Aug, its west edge retreats eastward away from the continent.





Separate indices have been used to define the intensity and location of the BH
(Stahle and Cleaveland, 1992; Ortegren et al. 2011, Li et al. 2011; Zhu and Liang,
2013). For the BH location, we adopted the BH longitudinal index (BH-Lon) from Li et
al. (2011) as a measure of the westward extension of the BH. They defined the
BH-Lon as the longitude of the cross point of the 1560 geopotential meter (gpm)
isoline and the 850hPa wind ridgeline. The BH-Lon is always negative in longitude,
thus more negative values meaning closer proximity to the US.
The intra-seasonal variability of the BH (c.f. Fig 3) will affect the value of BH-Lon
when defined with reference to a fixed isoline. For instance, the 1560 gpm isoline is
located further away from the HGB in August when the BH is weaker than in July,
even when the center location of the BH from the HGB does not change between the
two months. To reduce such effect, we tried different isolines with an interval of 4
gpm from 1560 to 1536 gpm in the calculation of the BH-Lon by month and
evaluated which definition captures the most variability of the HGB-mean ozone. For
June and July, the BH-Lon is best defined on the basis of the 1560 gpm isoline, the
same as in Li et al. (2011), and this definition results in the interannual (1993-2012)
correlation (r) of 0.69 between detrended BH-Lon and HGB-mean ozone. The BH-Lon
for August is defined using the 1556 gpm and the corresponding r is 0.62.
The time series of monthly-mean BH-Lon is shown in Figure 4. In
correspondence with the intra-seasonal movement of the BH shown in Fig 3, the
mean value of BH-Lon from 1990-2015 was 80.3°W in June, 93.1°W in July, and
91.6°W in August. There were no significant trends in BH-Lon for the months of June





and July. In August, however, BH-Lon exhibited a significant increasing trend (i.e., an
eastward shift) of 0.51° $a^{-1}$ (p<0.1) over 1990-2015. Shen et al. (2015) found an
increasing trend of 0.35° $a^{-1}$ in the JJA-mean BH-Lon over 1980 - 2010 using the
definition of Li et al. (2011). Rather than a change in the BH circulation patterns, they
attributed this trend to a spatially uniform decrease of SLP over the US and adjacent
Atlantic Ocean in the reanalysis data. Given their work as well as the lack of a
consistent trend in the monthly-mean BH-Lon over 1990-2015, we do not consider
the trend of BH-Lon in the present work and the BH-Lon time series were de-trended
by removing the 7-year moving averages.

Another type of BH index is defined on the basis of pressure differences

between two representative locations, with their exact locations varying among
studies. Zhu and Liang (2013) defined a pressure-based BH index (BHI) as the mean
SLP difference between a location in the Gulf of Mexico and the other in the
southern Great Plains where the SLP has the largest positive and negative correlation
with LLJ respectively. As a result, their BHI exhibits a significant positive association
between the strength of LLJ, which determines the transport of clean marine air
from the Gulf of Mexico. Similar to that study, we defined a pressure-based BHI as
the mean SLP difference along the west edge of the BH, between the same location
in the Gulf of Mexico as selected by Zhu and Liang (2013) (25.3°-29.3°N,
92.5°-87.5°W) (box 1 in Figure 1a) and the other in southern Great Plains (35°-39°N,
105.5°-100°W) (box 2 in Figure 1a) where the SLP exhibits the largest correlation
with the HGB-mean ozone. This BHI shows only weak to moderate correlations with





BH-Lon, with *r* being -0.57, -0.26 and -0.39 for JJA respectively, suggesting the
position of the BH west edge may not vary coherently with the pressure gradient
over the west edge. Therefore, both BH-Lon and BHI are used as predictors in the
regression model described below.
**2.4.  Statistical Method**

We applied a multiple linear regression (MLR) model, which has been

commonly used in air quality and meteorological studies (e.g. Kutner et al., 2004; Tai
et al., 2010), to construct the statistical relationship between the detrended
HGB-mean ozone and the five meteorological predictors described above, including
BH-Lon, BHI, T, U, and V. Since our focus is on variability, all the meteorological
predictors were detrended using the same approach as the ozone data. For ease of
comparison, all the meteorological predictors in the regression analysis were
normalized. The regression was conducted on the monthly scale from 1993 to 2012.
The model is of the form:
$$y = \beta_0 + \sum_{k=1}^{n} \beta_k x_k \tag{1}$$
where y is the detrended monthly HGB-mean MDA8 $O_3$, n is the number of
predictors, $x_k$ represents the $k^{th}$ predictor which is detrended and normalized, $\beta_k$
is the corresponding regression coefficient for $x_k$, and $\beta_0$ is the intercept. The
predictors are separated into two groups. We applied a stepwise regression using
the BH indices (BH-Lon, BHI) first, which represent the large-scale effects. The
HGB-mean T, U, and V that characterize local meteorological conditions were added



subsequently only if they result in significant improvements in model performance.
The predictor selection is based on the Akaike Information Criterion (AIC) statistics to
obtain the best model fit (Venables and Ripley, 2003).

**3.   RESULTS**
**3.1.   Ozone and the BH relationship**

The HGB-mean ozone shows a large intra-seasonal variation during JJA (Figure

5), with a minimum of monthly-mean ozone in July. We first examined if this feature
can be explained by near-surface temperature, which has been suggested as an
important meteorological factor affecting surface ozone in many regions (Fu et al.,
2015; Rasmussen et al., 2012; Camalier et al., 2007). As shown in Figure 5, the
HGB-mean temperature is the highest in July when the monthly mean ozone is
lowest, precluding temperature as the driver of the intra-summer variability of
ozone over the HGB region. This can be explained by the fact that summertime
temperatures over the HGB region are always high, thus the temperature variation is
relatively less significant and the ozone formation might not be limited by
temperature over this region in summer. To further support this argument, on the
interannual time scale (1993-2012) the HGB-mean ozone shows no correlation with
temperature in June (r = -0.14), although the correlation coefficient between the two
increases to 0.29 and 0.41 in July and August, respectively.

The BH-Lon reaches its westernmost location in July (Figure 3), coincident with





the ozone minimum (monthly mean) in the same month. This coincidence supports
the mechanism that the more westward shift of the BH (the lower BH-Lon) brings
the stronger inflow of cleaner maritime air into the HGB leading to lower surface
ozone. The decrease of background ozone over the HGB region from spring to
summer is reported by a number of observational and modeling studies
(Nielsen-Gammon et al., 2005a; Nielsen-Gammon et al., 2005b; Li et al., 2002;
Reidmiller et al., 2009) and can be attributed to strengthening of this maritime
inflow in summer. Indeed, as shown in Figure 6a, the BH-Lon shows a significantly
stronger correlation with the HGB-mean ozone during JJA (r = 0.62~0.69) than
temperature does. However, the BH-Lon does not correlate well with the HGB-mean
temperature, with r being -0.04, 0.44, and 0.33 for JJA respectively. The BH-Lon has a
higher correlation with the HGB-mean meridional wind (V) (r = -0.4~-0.7) but does
not correlate with the zonal wind (U). This is expected because the BH determines
the strength of the meridional flows that bring maritime air masses from the Gulf of
Mexico to southeastern Texas.
Given the intra-summer variations described above, the effects of the BH on the
HGB-mean ozone are analyzed month by month in the following sections. For
comparison, Zhu and Liang (2013) combined the intra-seasonal and interannal
variations of ozone and Shen et al. (2015) focused on the variability of the JJA-mean
ozone. Both studies investigated the southern US as a whole.
**3.2.  Statistical model**
Using the stepwise regression (Equation 1), we obtained the best-fit MLR



equations for the interannual variability of the HGB-mean ozone by month. Table 1
summarizes the regression results, including the predictors selected for each month
and their regression coefficients. Both BH-Lon and BHI are selected as significant
predictors ($p < 0.05$) for each month, while V is selected as an additional predictor
for August only. Temperature is not selected as a predictor by the MLR for any
month, which is consistent with insignificant correlation between ozone and
temperature presented above. The HGB-mean zonal wind (U) is not selected for any
month either. The squares of the Pearson correlation coefficients ($R^2$) from the MLR
are 0.61 for Jun, 0.72 for Jul, and 0.70 for Aug.

Figure 7 displays the time series of the observed and MLR-regressed

monthly-mean ozone (both detrended) from 1993 to 2012. Many of the extremely
high and low ozone months are well reproduced by the MLR model, for example, the
low ozone month of June 2004 and high ozone month of August 2011. The BH-Lon is
the most important predictor throughout JJA and has the highest regression
coefficient (absolute values) that is positive throughout JJA (Table 1). Given the
BH-Lon being negative, this means that as the Bermuda High extends more
westward, ozone over the HGB is lowered, which supports the mechanism that the
BH brings maritime inflow of low ozone background into the HGB region. The MLR
relationship indicates that monthly-mean MDA8 ozone over the HGB area will
decrease by 4.52, 3.44, and 3.34 ppbv, respectively, for every degree of westward
extension of the BH-Lon in June, July, and August.

While the MLR model developed here captures 61%-72% (based on $R^2$) of the



interannual variance of the HGB-mean ozone from June to August, the
meteorological predictors may be correlated with each other, leading to overfitting
of the data. To examine the multi-collinearity between the meteorological predictors,
the variance inflation factor (VIF), a widely used index of the collinearity in the
regression analysis, was calculated for each variable. Table 2 summarizes the VIF of
each predictor by month. Most of the VIF is smaller than 3, much lower than the
commonly-used VIF threshold of 10 in determining significant collinearity (Kutner et
al., 2004), indicating the problem of multi-collinearity among the predictors is
generally unimportant.
**3.3.  Model cross-validation**

The MLR model described above shows good regression performance in

explaining the interannual variations of the summertime HGB-mean ozone on the
monthly scale. To evaluate the predictability of this model, a cross-validation (CV)
method was implemented. We first isolated one year at a time, performed model
fitting with the remaining years, and then applied the model to predict the
monthly-mean ozone on the isolated year. Figure S1 shows the CV results by month.
The $R^2$ between the observed and CV-predicted ozone is 0.48~0.59, indicating the
MLR model is capable of predicting about 50% interannual (1993-2012) variability of
monthly mean ozone over the HGB area in summer. However, some of the extreme
ozone values are not well predicted, suggesting that other factors are responsible for
those high ozone events, e.g. emissions and stagnation conditions, which are not
considered as predictors in the MLR model.






## 4.  DISCUSSION

### 4.1.  Comparison with other studies

To the best of our knowledge, the MLR-regression correlation coefficients (r = 0.78~0.84) are significantly higher than those from previously published studies on the regression relationship between interannual variability of meteorological factors and surface ozone over a metropolitan region or the southern US as a whole. Zhu and Liang (2013) reported a negative correlation of -0.5 ~ -0.7 between the BHI and summer-mean MDA8 ozone during 1993-2010. Shen et al. (2015) identified the polar jet, the Great Plains low level jet, and the BH as major synoptic-scale patterns influencing surface $O_3$ variability in the eastern US in summer, which in combination explain 53% (r = 0.73) of the interannual variance of summer-mean MDA8 ozone in South Central US during 1980-2010. These studies averaged surface ozone over a large geographical region and onto a seasonal mean and thus smoothed out some variability. By comparison, the present study investigates the interannual variability of monthly mean ozone over a smaller region (HGB) and extends to more recent time periods (1990-2015). With just three meteorological predictors (BH-Lon, BHI, and V), the MLR model developed here captures 61% - 72% of the interannual variance of the HGB-mean ozone during JJA. The MLR model developed here shows a good prediction skill with the CV $R^2$ higher than 0.48 for each month of JJA.

### 4.2.  Mechanism





Among the meteorological predictors examined here, indicators of the BH
location (BH-Lon) and strength (BHI) explain more interannual variability of the
summertime HGB-mean ozone than local meteorological factors (i.e. winds and
temperature), indicating the dominant role of large-scale circulation patterns
controlling ozone variability over this region in summer. The BH-Lon is the most
important predictor throughout JJA, which alone explains 38%-48% of the observed
interannual variability of monthly mean ozone over the HGB region. As discussed
above, the BH circulation patterns in summer are responsible for the inflow of low
ozone air masses from the Gulf of Mexico into the HGB region. The MLR results
suggest that the BH-Lon is a better predictor than local-scale meridional wind to
indicate the variability of this maritime inflow and consequently surface ozone.
As a further illustration of this mechanism, Figure 8 compares circulation
patterns and associated changes in the HGB-mean ozone between two
representative months, Jul 2000 and Jul 2001. The BH was not only stronger in Jul
2001 but also extended closer to the southeastern US during that month than Jul
2000. The BH-Lon was 88.7$^{\circ}$W in Jul 2001 as compared to that of 78.7$^{\circ}$W in Jul 2000,
indicating stronger maritime influence to HGB during the former month.
Correspondingly, surface ozone concentrations were significantly lower at all the
CAMS monitors during Jul 2001 than Jul 2000 (Fig 8). The HGB-mean ozone was 38.9
ppbv in Jul 2001, compared with 49.0 ppbv in Jul 2000. The number of ozone
nonattainment days was also lower in Jul 2001 (2 days) than Jul 2000 (6 days).
Despite different BH-Lon, the HGB-mean V happened to be the same during the two



months, both -2.1 m/s. This indicates that local-scale wind alone may not be
sufficient to indicate the origin of air masses because winds are affected by both
large-scale and local-scale factors.

Given the large-scale influence of the BH, the mechanism underlying the BH and

$O_3$ association over the HGB region, if correct, should apply for other urban regions
along the Gulf Coast which experience similar onshore maritime flow from the Gulf
of the Mexico in summer. To test this, we chose New Orleans in Louisiana
(29.8°-30.6°N, 89.9°-90.7°W), Mobile in Alabama (30.4°-30.8°N, 88.0°-88.2°W), and
Pensacola in Florida (30.3°-30.5°N, 87.2°-87.3°W), all of which are at about 30°
latitude, similar to HGB. Figure 6b-d presents the time series of surface ozone
(detrended and averaged over all the CAMS monitors) at these urban areas during
JJA along with BH-Lon. On interannual time scales (1993-2013), monthly mean ozone
concentrations at the three urban regions all exhibit significantly positive
correlations with the BH-Lon throughout JJA (r = 0.38~0.64), which is consistent with
the BH-$O_3$ association over HGB and thus confirms the large-scale impact of the BH
circulation patterns on surface ozone variability along the Gulf Coast. It is interesting
to note that the BH-$O_3$ correlation coefficients at HGB are highest among the four
urban regions along the Gulf Coast. This can be partly explained by the fact that HGB
has the largest quantity of ozone-precursor emissions among the four urban areas,
so that the intrusion of clean maritime air masses would cause a larger relative
reduction of ozone at HGB, and therefore the BH location and strength could cause
more ozone variability.


### 4.3. Threshold value of BH-Lon


Shen et al. (2015) identified a threshold value of 85.4°W for the JJA-mean
BH-Lon that marks two different associations of the BH-Lon with surface ozone in the
eastern US. When the BH-Lon is east of 85.4°W, westward movement of the BH-Lon
leads to a decrease of surface ozone; when it is west of that threshold, its westward
movement leads to an increase of ozone. We did not find such a threshold value of
BH-Lon for the monthly scale analysis presented above; the monthly-mean
relationship between BH-Lon and HGB-mean ozone is consistently linear throughout
the study period for each summer month (c.f. Fig 6 and Fig S1).
On the daily scale, however, we found evidence for a threshold of BH-Lon that
tends to create conducive or non-conducive conditions for high ozone days over the
HGB region. Figure 9a shows the variation of HGB-mean daily ozone anomaly as a
function of daily BH-Lon during July 2011, a high ozone month. The daily ozone
anomaly is calculated as the difference between daily and 30-day moving average of
ozone. The daily data shown in Fig 9a clearly display two populations, warranting a
threshold of BH-Lon to separate them. As a simplified attempt, a linear piecewise
fitting was applied to the daily data, using different segmentation points with varying
BH-Lon from 100° W to 80° W at an interval of 0.5°. The least regression error (sum
of the squared errors) was obtained when the segmentation point was set to 90° W,
which represents the threshold of BH-Lon for July 2011. The resulting regression
indicates that when the BH-Lon is east of 90°W, the westward extension of BH-Lon
tends to cause a decrease of HGB ozone; the opposite relationship holds when the




BH-Lon is west of that threshold (Fig 9).

To understand causes of the two regimes, Figure 9b-c compares circulation

patterns between two representative days, July 3 and July 10 2011, when the BH-Lon
is located to the west and east of its threshold respectively. The circulation patterns
on July 10 (9c), when the BH-Lon was located to the east of 90°W, are similar to the
typical monthly-mean patterns described before: clean maritime air masses flow
southeasterly to southeast Texas along the west edge of the BH, leading to lower
surface ozone in HGB (26.4 ppbv on July 10 2011). On July 3 (9b), there was a
high-pressure system over southeast US, which appears to be separate from the BH
and in this case the choice of 1560 gpm isoline may not be appropriate to define the
BH west edge on a daily scale. According to the wind fields, however, this
high-pressure system over land is likely resulted from the westward extension of the
BH to the continental US. The high-pressure system typically brings stagnant weather,
high temperatures, and clear skies, which are all favorable meteorological conditions
leading to higher ozone on July 3 (42.2 ppbv).

To predict daily ozone on the basis of its statistical relationships with

meteorology alone is a challenging task and beyond the scope of the present study.
The simple daily analysis presented above for the month of July 2011 was intended
to demonstrate that the linear relationship between the BH-Lon and HGB-mean
ozone on a monthly scale is useful to explain some portion of HGB ozone variability
on a daily scale, but there are certain degrees of nonlinearity in the daily relationship
associated with a threshold value of the BH-Lon that separates different circulation




regimes and that threshold value may not be constant for every month. When the
BH-Lon is used to predict the monthly-mean ozone over the HGB area, however, it is
not necessary to consider the threshold of BH-Lon since the mean variations of the
BH-Lon is a lot smaller on the monthly scale than that on the daily time scale.

**5. Conclusion**

More than two decades (1990 - 2015) long observational record of MDA8 ozone

and meteorology were analyzed to characterize the effects of the BH circulation
patterns on interannual variations of surface ozone in the HGB region during the
summer months (June to August). The BH indicators are the longitude of the BH
western edge (BH-Lon) and the pressure-based BH intensity index (BHI) along its
western edge. Statistical relationships between the HGB-mean ozone variability and
meteorological predictors, including both large-scale (BH-Lon, BHI) and local-scale
ones (T, U, V), were tested and developed through multiple linear regression (MLR).
The best-fit MLR equations select both BH-Lon and BHI as significant predictors ($p <$
$0.05$) of interannual variability of the HGB-mean ozone for each month and
meridional wind speed (V) as a significant predictor for August only. Temperature or
zonal wind speed (U) is not selected as a predictor by the MLR for any of the summer
month. The exclusion of temperature in the MLR model is supported by the lack of
significant correlations between ozone and temperature over the HGB region on
both intra-seasonal and interannual time scales. This suggests temperature is not a
key driver of summertime ozone variability in the HGB region despite its importance



for other regions.
With only three meteorological predictors (BH-Lon, BHI, and V), the MLR model
developed here captures 61%-72% (based on $R^2$) of the interannual variance of the
HGB-mean ozone from June to August. The MLR model developed here also show a
good prediction skill with the CV $R^2$ higher than 0.45. The BH-Lon alone explains
38%-48% (r = 0.62~ 0.69) of the year-to-year variability in monthly mean ozone over
HGB during JJA for the period 1990-2015, indicating the dominant role of large-scale
circulation patterns controlling ozone variability over this region in summer. Such a
high correlation is explained by the mechanism that the western extension of the BH
determines the strength of the inflow of maritime air masses with lower ozone
background from the Gulf of Mexico to the HGB during summer. This mechanism
also applies to other coastal urban regions, such as New Orleans LA, Mobile AL and
Pensacola FL, confirming the large-scale impact of the BH circulation patterns on
surface ozone variability along the Gulf Coast. The linear relationship between the
BH-Lon and HGB-mean ozone on a monthly scale is useful to explain some portion of
HGB ozone variability on a daily scale, but there are certain degrees of nonlinearity
in the daily relationship associated with a threshold value of the BH-Lon that
separates different circulation regimes and that threshold value may not be constant
for every month. The statistical relationship between surface ozone and large-scale
circulation patterns on derived herein will be useful to distinguish the role of
meteorology versus anthropogenic emissions in controlling the interannual
variability of ozone along the Gulf Coast as well as to serve a benchmark to test the





performance of air quality models in representing such distinctions, which will be
investigated in future studies.

**Acknowledgement**
This research was supported by Texas Commission on Environmental Quality
(TCEQ) (Grant No. 582-13-34576) and Texas Air Quality Program (Project 14-010). B.
Jia acknowledges additional funding from the National Key Basic Research Program
of China (2013CB956603).

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



**Tables**

Table 1. Regression coefficients and coefficients of determination ($R^2$) of the best-fit

MLR models for June, July, and August.   The model cross-validation $R^2$ is shown in

the last column.

| | $\beta_0$ (intercept) | BH-Lon | BHI | V | $R^2$ | Adjusted $R^2$ | Cross-validation $R^2$ |
|---|---|---|---|---|---|---|---|
| June | 0.16 | 4.52 | -3.84 | - | 0.61 | 0.56 | 0.48 |
| July | -0.26 | 3.44 | -2.98 | - | 0.72 | 0.69 | 0.59 |
| August | -0.14 | 3.34 | 3.21 | -5.21 | 0.70 | 0.64 | 0.51 |

Table 2. Variance inflation factor (VIF) of the predictors selected in the MLR mode for

June, July and August.

| | BH-Lon | BHI | V |
|---|---|---|---|
| June | 1.47 | 1.47 | - |
| July | 1.07 | 1.07 | - |
| August | 1.21 | 2.21 | 2.22 |



612                                          **Figures**

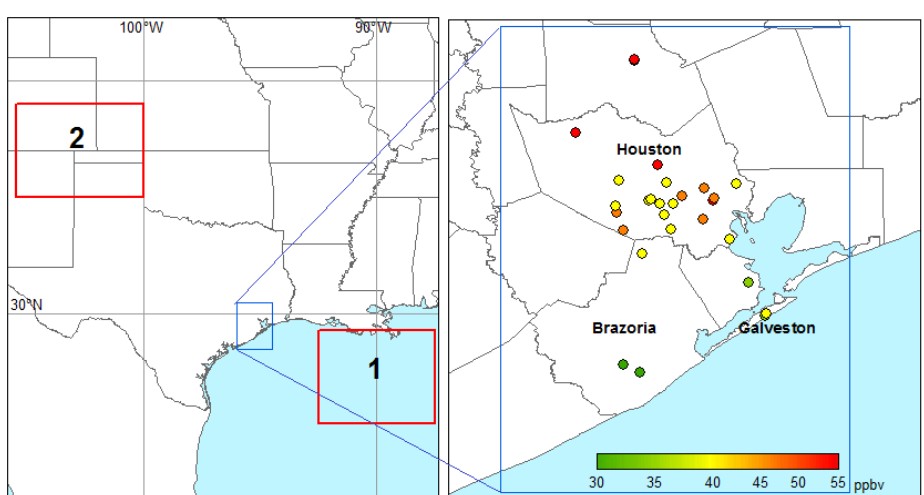


Figure 1. (a) Locations of the HGB region (blue box). The red boxes show the regions
used to define the BH intensity indices BHI; (b) Locations of the CAMS sites within
the HGB region and the long-term (1990-2015) mean MDA8 ozone from June to
August.

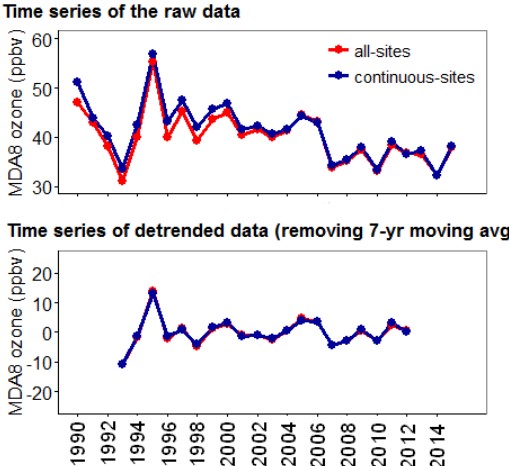


Figure 2. Summer (JJA)-mean ozone over the HGB area during the period from 1990
to 2015. The upper panel is the time series of the raw data and the lower panel is the





detrended time series after subtracting the 7-year moving averages. The red line is
the average from all available sites for each year and the blue line is the average of
those sites with continuous coverage from 1990 to 2015.

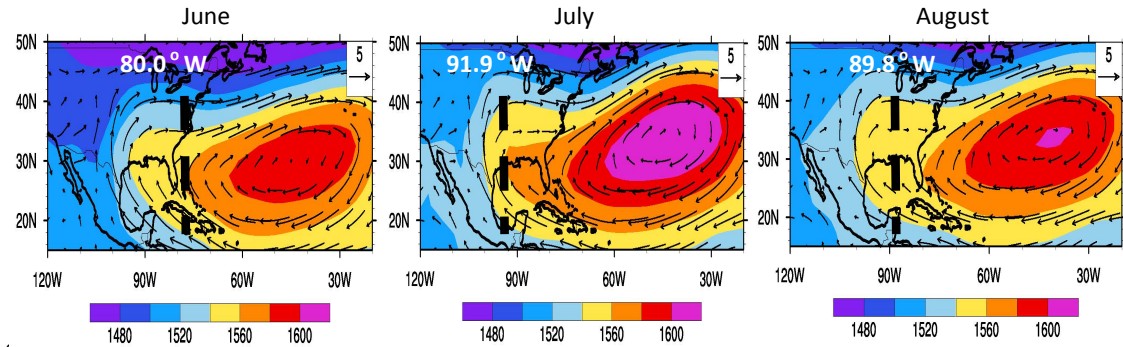

Figure 3. Distributions of the 1998-2013 mean 850hPa geopotential height (color
contour) and wind fields (arrows) in June (left), July (center), and August (right). The
black dashed line shows the longitude of the BH-Lon (values shown in white).



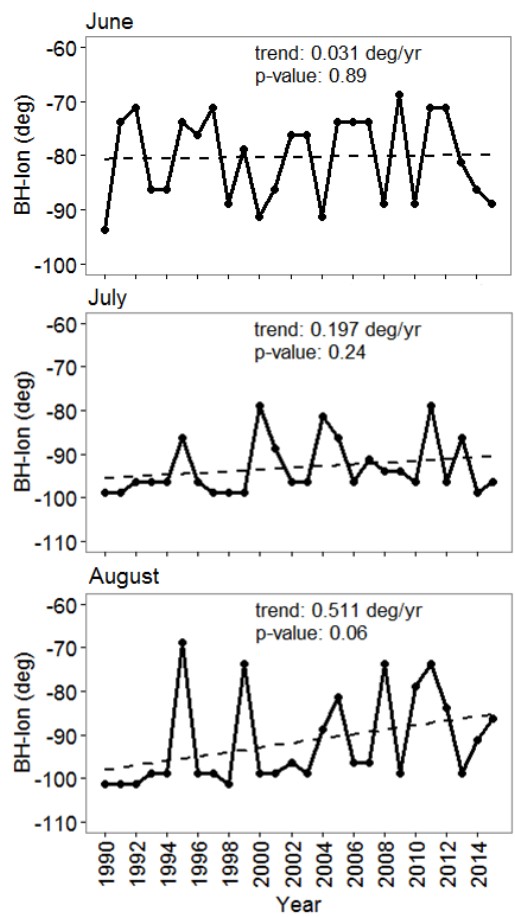


Figure 4. Time series of the BH-Lon (solid line) in June, July and August from 1990 to
2015.    A linear trend (dash line) is used to fit each time series.





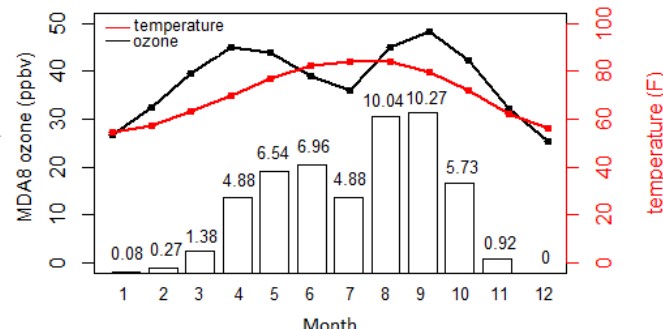


Figure 5. The 1990-2015 mean monthly MDA8 ozone over the HGB area (black line),
overlaid with the average number of exceedance days by month (bars) and the
HGB-mean temperature (red line). An exceedance day is defined when MDA8 ozone
at one or more monitors in HGB is higher than 75 ppbv.








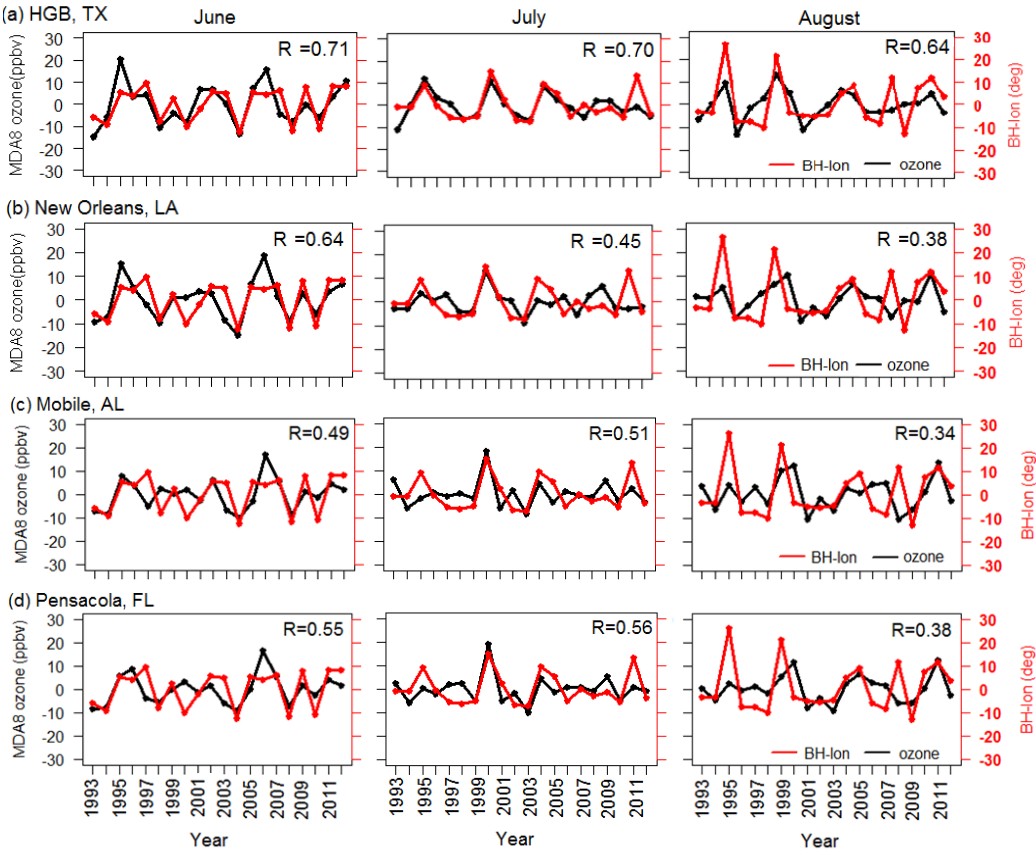

Figure 6. The time series of detrended BH-Lon (red line) and MDA8 ozone (black line)
during June, July and August over the HGB region, TX (a), New Orleans, LA (b),
Mobile, AL (c) and Pensacola, FL (d). The correlation coefficients between BH-Lon
and ozone are shown in each figure.







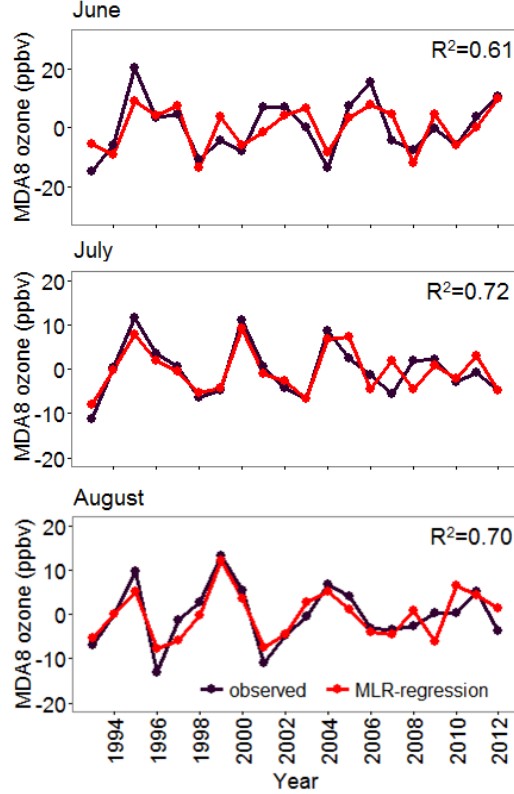


Figure 7. Time series of observed HGB-mean MDA8 ozone (black line) and
MLR-regressed ozone (red line) in June (top), July (middle) and August (bottom). The
data presented are detrended.










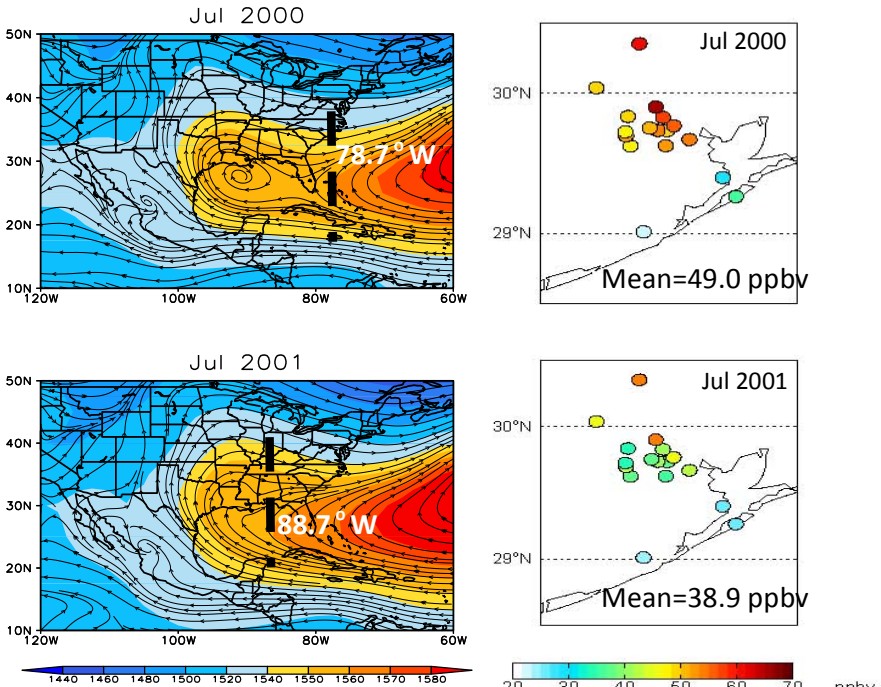


Figure 8. (Left) 850 hPa geopotential height (gpm) and stream line for the month f
July 2000 and July 2001. The black dashed line shows the longitude of the BH-Lon
(values shown in white). (Right) MDA8 ozone concentrations at the CAMS sites in
HGB for July 2000 and July 2001.







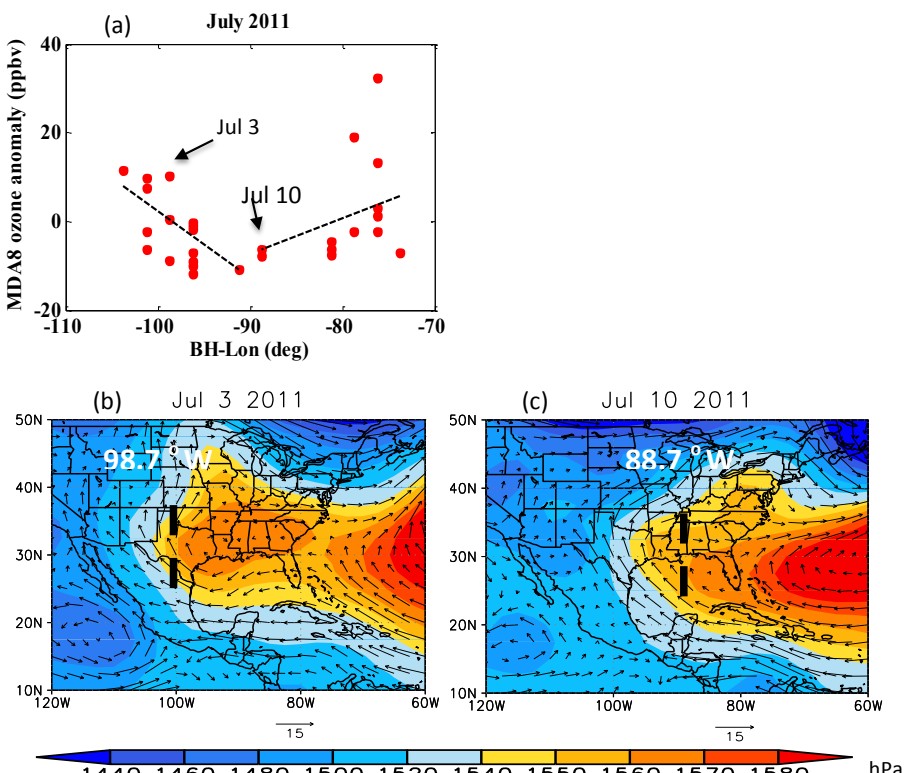


Figure 9. (a) Relationship between daily ozone anomaly (y-axis) and daily BH-Lon

(y-axis) for July 2011. (b-c) Distribution of SLP (color contours) and 850hPa wind

fields (arrows) on July 3rd (b) and July 10th 2011 (c). The black dashed line shows the

longitude of the BH-Lon (values shown in white).



