# Peer review of "Influence of the Bermuda High on interannual variability of"

_Atmospheric Chemistry and Physics, 2016_

## Referee Comment (RC1) · Anonymous Referee #1 · 3 Oct 2016

Inspired by previous research on the impact of the Bermuda High on surface ozone in Eastern and Southern US, the authors constructed an MLR model to quantitatively describe the relationship between MDA8 ozone in the Houston-Galveston-Brazoria region and the Bermuda High, using an intensity index, BHI, and a location index, BH-Lon. The analysis found that the Bermuda indices, in particular, BH-Lon, are better predictors for ozone prediction than local meteorological parameters such as temperature. The authors suggested that underlying mechanism is that the location and intensity of the BH control whether low-ozone maritime air from the Gulf can enter the HGB region, and the method may apply to other coastal regions along the Gulf coast. The paper is well written and provides good insights into the topic. I recommend the paper for

publication if the following comments are addressed.

Major comments

1. I am concerned that the de-trending method for BH-Lon is not robust and add noise to the regression analysis. I also suspect that the V wind may not be necessary for August if there weren't the statistical noise induced by the de-trending method.

Fig. 4 shows that BH-Lon has large inter-annual variations and tends to cluster in two groups. This feature is most significant in August, where the two groups of data are distant apart from each other. One group is around -70o (eastern group) and the other around -100 o (western group). As a result, the 7-year moving averages may be very noisy, depending on the specific time series. For example, in both Year 1997 and 2000, BH-Lon are -100 o, and we expect the values for these two years are close after de-trending. However, after removing the 7-year moving averages, they become -7o and -3 o (calculated based on the figure), respectively. The 4 o difference (quite significant according to the regression coefficient) between the two years results mainly from the variation in the 7-year averages. Notice that the 7-year moving average for 1997 includes two "eastern" years (1995 and 1999), while that for 2000 only includes one (1999). This kind of noise induced by the de-trending method may degrade the explanatory power of BH-Lon, especially in August.

In my opinion, it is a better method to just use the raw data because the trend is not significant in the first place. Even for August, I suspect that the trend is mainly from the first 4 years. Should the first 4 years are removed or a non-parametric regression method is used, the trend for August might well also be insignificant (p>0.1).

Minor comments

1. Line 195-197. The authors used a southern Great Plains domain that is different from the definition in Zhu and Liang (2013). Are these two domains very different?

2. Line 255-256. The logic here is a little confusing because the previous paragraph

seems to suggest that BH-Lon is able to capture the intra-season variation pretty well (at least much better than temperature does), so what "described above" does not support the "month by month" analysis. That said, I am fine with the "month by month" analysis.

3. Line 269. "The squares of the Pearson correlation coefficients (R2)" should be changed to "The coefficients of determination (R2)" because the latter is the correct term in this context and is also consistent with the caption of Table 1.

4. Figure 8. Fig. 8 used the stream line. But these stream lines were not discussed in the text. I suggest to use wind vectors rather than stream lines, for 1) the consistence with other Figures (Fig.3 and 9); 2) The V wind speed for the two months is mentioned in the text (Line 348-349) and the information can be better visually shown with wind vectors.
* * *

---

## Referee Comment (RC2) · Anonymous Referee #2 · 10 Oct 2016

This manuscript examines whether interannual variability in summertime monthly ozone concentrations in Houston can be explained by the strength and location of the Bermuda High. Through multiple linear regression analysis, the authors show that a remarkable degree of ozone variability can be explained by the intensity and longitudinal extent of the Bermuda High. These features of the large scale circulation patterns can explain even more of the interannual variability than local temperatures or winds. It is useful that the authors briefly touch on the influence of BH metrics on ozone in other Gulf Coast cities, to show the extent to which the conclusions for Houston might apply elsewhere.

The methods of the paper are sound and its findings are well explained. The

manuscript merits publication in ACP after addressing the minor comments noted below.

1. The paper focuses on June, July and August, noting in Lines 100-101 that this is when the Bermuda High is closer to North America and more influential on circulation patterns over Houston. However, as shown in Figure 5, Houston ozone exhibits a bimodal seasonality, with some of the highest ozone and exceedance rates occurring in the spring and early fall rather than in JJA. If the meteorological features identified here are unable to predict peak ozone outside of JJA, this should be noted as a limitation of the study.

2. The ozone standard is now 70 ppb, though the paper uses the earlier 75 ppb standard as the exceedance threshold.

3. It should be clarified in Lines 92-93 how the Bermuda High influences nocturnal low level jets.

4. Meteorological data is taken from a 2.5 x 2.5 degree reanalysis, but the longitude of the Bermuda High is reported with 0.1 degree precision. Clarify how BH-Lon was computed from the data.

5. Line 159: Clarify what is meant by the 850hPa wind ridgeline.

6. The authors choose to de-trend the Bermuda High longitude data, though the reasons behind the trend remain unclear (lines 176-184). It would be helpful to note how the results would have been affected if BH-Lon had not been de-trended.

7. It is unclear how Figure 6a illustrates the claim in lines 247-249.

8. Where was the correlation observed in Zhu and Liang (lines 312-314)

9. In Figure 3, I don't see the black dashed line, and the units of the "5" arrow should be clarified.

Minor technical corrections: Line 63: replace "the high pressures" with "high pressure";

Line 160: replace "the US" with "Houston"; Line 343: replace "the former month" with "2001" for clarity.

---

## Author Comment (AC1) · 11 Nov 2016

**Response to Reviews**

We thank both reviewers for their constructive comments to improve the manuscript. Their comments are reproduced below with our responses in blue. The corresponding changes in the manuscript are highlighted in blue.

**Reviewer #1**

Inspired by previous research on the impact of the Bermuda High on surface ozone in Eastern and Southern US, the authors constructed an MLR model to quantitatively describe the relationship between MDA8 ozone in the Houston-Galveston-Brazoria region and the Bermuda High, using an intensity index, BHI, and a location index, BH-Lon. The analysis found that the Bermuda indices, in particular, BH-Lon, are better predictors for ozone prediction than local meteorological parameters such as temperature. The authors suggested that underlying mechanism is that the location and intensity of the BH control whether low-ozone maritime air from the Gulf can enter the HGB region, and the method may apply to other coastal regions along the Gulf coast. The paper is well written and provides good insights into the topic. I recommend the paper for publication if the following comments are addressed.

Major comments
1. I am concerned that the de-trending method for BH-Lon is not robust and add noise to the regression analysis. I also suspect that the V wind may not be necessary for August if there weren't the statistical noise induced by the de-trending method.

Fig. 4 shows that BH-Lon has large inter-annual variations and tends to cluster in two groups. This feature is most significant in August, where the two groups of data are distant apart from each other. One group is around $-70^{o}$ (eastern group) and the other around $-100^{o}$ (western group). As a result, the 7-year moving averages may be very noisy, depending on the specific time series. For example, in both Year 1997 and 2000, BH-Lon are $-100^{o}$, and we expect the values for these two years are close after de-trending. However, after removing the 7-year moving averages, they become $-7^{o}$ and $-3^{o}$ (calculated based on the figure), respectively. The $4^{o}$ difference (quite significant according to the regression coefficient) between the two years results mainly from the variation in the 7-year averages. Notice that the 7-year moving average for 1997 includes two "eastern" years (1995 and 1999), while that for 2000 only includes one (1999). This kind of noise induced by the de-trending method may degrade the explanatory power of BH-Lon, especially in August.

In my opinion, it is a better method to just use the raw data because the trend is not significant in the first place. Even for August, I suspect that the trend is mainly from the first 4 years. Should the first 4 years are removed or a non-parametric regression method is used, the trend for August might well also be insignificant (p>0.1).

We agree with the reviewer's concern and suggestion. To address them, we have conducted additional analysis, as presented below, and verified that our results are not affected by different detrending methods.

The following table lists the regression coefficient of determination ($R^2$) using different treatments of BH-Lon and $O_3$. Note $O_3$ is the dependent variable and BH-Lon the independent variable, thus we favor the model settings in which both data are processed the same way. The 1st setting (raw data for both) has low $R^2$ for July and Aug when HGB ozone has a significant decreasing trend. Such trend is driven by decreasing emissions, a factor not considered in the MLR (hence poor model performance). The 2nd setting uses raw data for BH-Lon, as suggested by the reviewer, and this setting has $R^2$ values only slightly less than what we reported in the manuscript (i.e. the 4th setting). The 3rd setting uses the simple linear regression to remove trends in both BH-Lon and $O_3$. This yields $R^2$ higher than 0.5 for all the months, although 10-20% lower than the 4th setting.

| BH-Lon and $O_3$ in MLR | June | July | August |
|---|---|---|---|
| (1) Raw data for both. | 0.52 | 0.35 | 0.24 |
| (2) Raw data for BH-Lon only; $O_3$ is de-trended by subtracting the 7-yr moving average from the raw data. | 0.60 | 0.67 | 0.65 |
| (3) Both de-trended by removing the linear trend from the raw data. | 0.52 | 0.58 | 0.56 |
| *(4) Both de-trended by subtracting the 7-yr moving average from the raw data. | 0.61 | 0.72 | 0.70 |

*This *setting is reported in Table 1 of the manuscript.*

We have added the above table in the Supplementary Material (Table S2) and added more discussion on this issue in the main text (line 283-289). We chose to report the 4th setting in the main text based on the consideration of four factors: (1) the BH-Lon and $O_3$ (and other variables) should be processed in a consistent manner in the MLR model; (2) $O_3$ data needs to be de-trended to remove the influence of changing anthropogenic emissions; (3) we do not know if the ozone trend is linear during the timeframe of 1990-2015, thus the 7-yr moving average appears to be a better choice to de-trend ozone data; and (4) a number of previous publications which analyze the effects of meteorology on air quality variability de-trend the data by subtracting the moving average from the raw data (e.g. Shen et al., 2015; Tai et al., 2010) and thus our results can be directly compared with them.

A remaining question is whether we should include V wind into the regression model for August. We found that the model performance would be significantly degraded if V wind was removed, regardless of which de-trending methods were used. This suggests the effect of V wind in the MLR is not random or resulted from noises, but robust. The table below shows the MLR $R^2$ for the model in August if V wind is

removed.

| BH-Lon and $O_3$ in August MLR model | Without V | With V |
|---|---|---|
| (1) Raw data for both. | 0.08 | 0.24 |
| (2) Raw data for BH-Lon only; $O_3$ is de-trended by subtracting the 7-yr moving average from the raw data. | 0.32 | 0.65 |
| (3) Both de-trended by removing the linear trend from the raw data. | 0.20 | 0.56 |
| *(4) Both de-trended by subtracting the 7-yr moving average from the raw data. | 0.41 | 0.70 |

*This *setting is reported in Table 1 of the manuscript.*

We have added the table in the supplementary material (Table S3), and clarified the main text (line 289-292). In addition, following the suggestion by the reviewer, we have added that for August the trend of BH-Lon would become insignificant if the first four years were removed from the time series (line 183-184),

References:

Shen, L., L. J. Mickley, A. P. K. Tai: Influence of Synoptic Patterns on Surface Ozone Variability over the Eastern United States from 1980 to 2012, Atmos. Chem. Phys., 15, 13073-13108, 2015

Tai, A.P.K., L.J. Mickley, D.J. Jacob: Correlations between fine particulate matter (PM2.5) and meteorological variables in the United States: Implications for the sensitivity of PM2.5 to climate change, Atmos. Environ., 44, 3976-3984, 2010

Minor comments
1. Line 195-197. The authors used a southern Great Plains domain that is different from the definition in Zhu and Liang (2013). Are these two domains very different?
We have corrected this typo in our old manuscript. The southern Great Plains domain is same as the definition in Zhu and Liang (2013), while the Gulf of Mexico domain is slightly different. Our Gulf of Mexico domain is located 2.5$^o$ east from the one in Zhu and Liang (2013), and this is because our domain was selected based on the correlation of SLP with the HGB-mean ozone, while their domain was based on the correlation of SLP with low-level jet. We have clarified in the text:

Line 201-205. "…we defined a pressure-based BHI as the mean SLP difference along the west edge of the BH, between the same location in southern Great Plains as selected by Zhu and Liang (2013) (35°-39°N, 105.5°-100°W) (box 2 in Figure 1a) and the other in the Gulf of Mexico (25.3°-29.3°N, 92.5°-87.5°W) (box 1 in Figure 1a) where the SLP exhibits the largest correlation with the HGB-mean ozone. Our Gulf of Mexico domain is located 2.5$^o$ east of that defined by Zhu and Liang (2013)."

2. Line 255-256. The logic here is a little confusing because the previous paragraph

seems to suggest that BH-Lon is able to capture the intra-season variation pretty well (at least much better than temperature does), so what "described above" does not support the "month by month" analysis. That said, I am fine with the "month by month" analysis.

Thanks for making this good point. We have clarified in the text.

Line 263. "Considering the large intra-summer variation in ozone, BH and their association, the effects of the BH on the HGB-mean ozone are analyzed month by month …".

3. Line 269. "The squares of the Pearson correlation coefficients (R2)" should be changed to "The coefficients of determination (R2)" because the latter is the correct term in this context and is also consistent with the caption of Table 1.

Done.

4. Figure 8. Fig. 8 used the stream line. But these stream lines were not discussed in the text. I suggest to use wind vectors rather than stream lines, for 1) the consistence with other Figures (Fig.3 and 9); 2) The V wind speed for the two months is mentioned in the text (Line 348-349) and the information can be better visually shown with wind vectors.

This is an excellent suggestion. We've replaced the stream lines in Fig 8 with wind vectors.

---

## Author Comment (AC2) · 11 Nov 2016

**Response to Reviews**

We thank both reviewers for their constructive comments to improve the manuscript. Their comments are reproduced below with our responses in blue. The corresponding changes in the manuscript are highlighted in blue.

**Reviewer 2**

This manuscript examines whether interannual variability in summertime monthly ozone concentrations in Houston can be explained by the strength and location of the Bermuda High. Through multiple linear regression analysis, the authors show that a remarkable degree of ozone variability can be explained by the intensity and longitudinal extent of the Bermuda High. These features of the large scale circulation patterns can explain even more of the interannual variability than local temperatures or winds. It is useful that the authors briefly touch on the influence of BH metrics on ozone in other Gulf Coast cities, to show the extent to which the conclusions for Houston might apply elsewhere. The methods of the paper are sound and its findings are well explained. The manuscript merits publication in ACP after addressing the minor comments noted below.

1. The paper focuses on June, July and August, noting in Lines 100-101 that this is when the Bermuda High is closer to North America and more influential on circulation patterns over Houston. However, as shown in Figure 5, Houston ozone exhibits a bimodal seasonality, with some of the highest ozone and exceedance rates occurring in the spring and early fall rather than in JJA. If the meteorological features identified here are unable to predict peak ozone outside of JJA, this should be noted as a limitation of the study.
We have stated this limitation. Line 268-271: "We note here that HGB ozone exhibits a bimodal seasonality (c.f. Figure 5), with 41% of exceedance days occurring in JJA, and the rest in the spring and early fall. The meteorological features identified here are not expected to predict peak ozone outside of JJA, which is a limitation of the study."

2. The ozone standard is now 70 ppb, though the paper uses the earlier 75 ppb standard as the exceedance threshold.
We have changed the threshold to 70 ppb and re-calculated the exceedance days. See the revised Figure 5.

3. It should be clarified in Lines 92-93 how the Bermuda High influences nocturnal low level jets.
Clarification has been added. Line 93-94: "…due to the superposition of the sea breeze cycle on a strong synoptic-scale southerly flow."

4. Meteorological data is taken from a 2.5 x 2.5 degree reanalysis, but the longitude

of the Bermuda High is reported with 0.1 degree precision. Clarify how BH-Lon was computed from the data.

BH-Lon is calculated as the longitude of the cross-point of the 1560 geopotential meter (gpm) isoline and the 850hPa wind ridgeline (defined below). The location of the 1560-gpm isoline is linearly interpolated to 0.1 degree precision within the 2.5 x 2.5 degree grid which contains this isoline. We have clarified this in the text (line 163-165).

5. Line 159: Clarify what is meant by the 850hPa wind ridgeline.

The ridgeline refers to the roughly zonal line that separates the easterly trade winds in the south from the westerly winds in the north at 850hPa. Mathematically it can be written as $u$=0 and $\frac{\partial u}{\partial y} \geq 0$, where $u$ is the zonal wind component and $y$ is meridional coordinate. We have clarified it in the text (line 160-163).

6. The authors choose to de-trend the Bermuda High longitude data, though the reasons behind the trend remain unclear (lines 176-184). It would be helpful to note how the results would have been affected if BH-Lon had not been de-trended.

We have clarified in the text that the main reason to de-trend the BH-Lon data is to be consistent with the treatment of ozone data in the MLR model (line 190-191). We have verified that the results are not affected if BH-Lon is not de-trended or de-trended with a different method. See the detailed explanation of this point in our response to the 1[st] reviewer and the added Table S2 and S3 in the supplementary material and related discussion in the text (line 283-292).

7. It is unclear how Figure 6a illustrates the claim in lines 247-249.

We've removed the reference of Figure 6a here and corrected the *r* values.

8. Where was the correlation observed in Zhu and Liang (lines 312-314)

It was over the southern Great Plains (including Houston). We've clarified this point. See line 336.

9. In Figure 3, I don't see the black dashed line, and the units of the "5" arrow should be clarified.

For clarity we now use the white dashed line to indicate the BH-Lon. The arrow indicates wind speed (m/s) and unit has been added.

Minor technical corrections:

Line 63: replace "the high pressures" with "high pressure";

Corrected.

Line 160: replace "the US" with "Houston";

Corrected.

Line 343: replace "the former month" with "2001" for clarity.
Corrected.